# Activity Pruning for Efficient Spiking Neural Networks

**Tong Bu**
Institution for Artificial Intelligence
School of Computer Science
Peking University
putong30@pku.edu.cn

**Xinyu Shi**
Institution for Artificial Intelligence
School of Computer Science
Peking University
xyshi@pku.edu.cn

**Zhaofei Yu** *
Institution for Artificial Intelligence
School of Computer Science
Peking University
yuzf12@pku.edu.cn

## Abstract

While sparse coding plays an important role in promoting the efficiency of biological neural systems, it has not been fully utilized by artificial models as the activation sparsity is not well suited to the current structure of deep networks. Spiking Neural Networks (SNNs), with their event-driven characteristics, offer a more natural platform for leveraging activation sparsity. In this work, we specifically target the reduction of neuronal activity, which directly leads to lower computational cost and facilitates efficient SNN deployment on Neuromorphic hardware. We begin by analyzing the limitations of existing activity regularization methods and identifying critical challenges in training sparse SNNs. To address these issues, we propose a modified neuron model, AT-LIF, coupled with a threshold adaptation technique that stabilizes training and effectively suppresses spike activity. Through extensive experiments on multiple datasets, we demonstrate that our approach achieves significant reductions in average firing rates and synaptic operations without sacrificing much accuracy. Furthermore, we show that our method complements weight-based pruning techniques and successfully trains an SNN with only 0.06 average firing rate and 2.22M parameters on ImageNet, highlighting its potential for building highly efficient and scalable SNN models. Code is available at https://github.com/putshua/Activity-Pruning-SNN.

## 1   Introduction

The human brain is remarkably energy-efficient, yet capable of performing a wide range of complex tasks, such as reasoning and planning. One of the key factors behind this efficiency lies in the behavior of the neurons. As a general strategy of the neural representation in the neural system [Yoshida and Ohki, 2020], the flexibility of sparse coding enables neurons to operate in a highly energy-efficient manner. For example, despite the large population of neurons in specific areas of the brain tasked with particular functions, only a subset of the neuron populations will be engaged simultaneously. Experimental evidence shows that sparse representation of visual sensory information in the primary visual cortex enhances the selectivity and sparseness of individual neurons, thereby improving the computational efficiency of the visual system [Vinje and Gallant, 2000].

---

*Corresponding author

39th Conference on Neural Information Processing Systems (NeurIPS 2025).

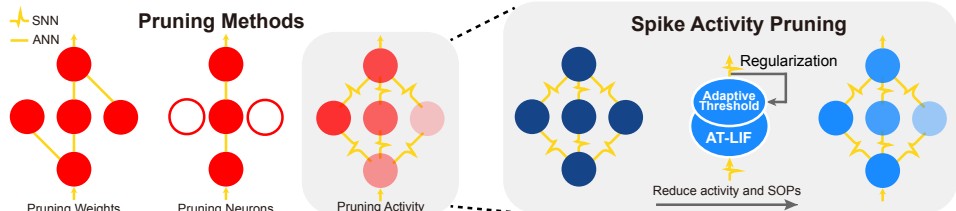

Figure 1: The left figure illustrates three pruning techniques for deep models. While all methods generally reduce model complexity, activity-based pruning is particularly effective for SNNs. The Right part shows the basic idea of our method, where the adaptive neurons (AT-LIF) are applied to regularize the threshold, so as to reduce the firing rate while stabilizing the training process.

A similar principle can be applied to artificial computational models through activation-based pruning, which seeks to moderate neuron activations to mirror the efficiency in sparse neural coding and gain efficiency in terms of time and energy consumption. However, activation pruning has not emerged as a mainstream approach for reducing network complexity compared to more impactful pruning methods that target network weights or structure [Kurtz et al., 2020], since the modest unstructured sparsity induced in activation maps seldom translates into meaningful efficiency gains with artificial neural networks on modern hardware. In contemporary architectures, the primary computational workload is carried out via matrix multiplications, where lowering a few activation values does not significantly cut down on computation time or resource usage.

Unlike traditional architectures, activity-based pruning offers a promising approach for brain-inspired models [Li et al., 2024], such as spiking neural networks (SNNs). SNNs are a well-established type of neural network that emulate the behavior of biological neurons and are increasingly recognized as a more efficient alternative to conventional computational models [Maass, 1997]. One important characteristic of the spiking neural networks is that it encodes information through discrete binary spikes. Therefore, the reduction of the neural activity in SNNs will directly lead to the sparsification of the output spike vector and the reduction of the firing rate, leading to potential computational efficiency. It is possible to create a highly sparse network in which only a few neurons will be engaged in one round of the computation with spike activity pruning, thus directly reducing the number of operations of the model. Thanks to the rapid recent advancements of neuromorphic hardware [Pei et al., 2019, DeBole et al., 2019, Davies et al., 2018, Nieves and Goodman, 2021, Fang et al., 2020, Zenke and Neftci, 2021, Yao et al., 2024], such sparse property can be effectively leveraged by specific neuromorphic hardware to work in an efficient and energy-saving manner, making these networks ideal for deployment in resource-constrained environments. Even on traditional architectures such as GPUs and deep learning accelerators, it is possible that such models can still benefit from activation sparsity to achieve memory savings and inference speedup. This approach enables the creation of models that are capable of performing complex tasks with minimal energy consumption, leading to more brain-like AI systems in terms of efficiency.

In this paper, we investigate an activity pruning technique aimed at suppressing the spike output. Our approach explores the capabilities of a novel neuron model in solving the learning dilemma in sparse models and stabilize the training process. We also introduce a threshold adaptation mechanism to effectively reduce neural activity. Our contributions can be summarized as:

- We specifically target the reduction of neuronal spike activity in SNNs, which directly lowers computational cost and serves as a unique advantage for efficient computation. To achieve this goal, we analyze the limitations of existing activity regularization methods and identify critical challenges associated with training sparse SNN models.

- We propose a novel neuron model, AT-LIF, designed to stabilize the training of sparse spiking networks while effectively suppressing spike activity. It incorporates a current-based output and an adaptive threshold mechanism, which help alleviate the gradient vanishing problem and resolve optimization conflicts.

- Extensive experiments demonstrate that our method significantly reduces spike activity and synaptic operations while maintaining performance comparable to the baseline methods. We further combine our proposed method with weight pruning strategies to illustrate its potential for achieving highly sparse SNNs.

## 2 Background

### 2.1 Related works

Network pruning is a well-established technique for artificial neural networks, with various approaches designed to reduce model complexity and improve efficiency. These pruning methods can typically be categorized into three types: unstructured weight pruning [Han et al., 2015, Zhu and Gupta, 2018, Kusupati et al., 2020], structured weight pruning [Wen et al., 2016, Mao et al., 2017, Anwar et al., 2017], and structured neuron pruning [Yu et al., 2018, Zhuang et al., 2020]. Similar exploration has been conducted with SNNs, where specific supervised learning algorithms are applied to prune weights or neurons [Wu et al., 2019, Yin et al., 2021]. Most distinguishing SNN pruning methods are inspired by biological principles, aiming to replicate the brain's efficient learning scheme to obtain sparse connected deep SNNs [Bellec et al., 2018, Chen et al., 2021, 2022, Shen et al., 2023]. The others draw inspiration from existing deep learning techniques and focus on jointly sparsifying the connections and neurons, introducing a complete pruning framework on deep SNNs [Deng et al., 2021, Shi et al., 2024]. Notably, while various pruning methods have been proposed for spiking neural networks Yin et al. [2021], Wu et al. [2019], Zhou et al. [2024], most focus on weight or neuron pruning rather than developing activity-based pruning approaches in depth. The majority of the existing pruning methods rely on adding regularization terms to constrain the firing rate of individual neurons [Deng et al., 2021, Yan et al., 2022], or regularizing the activation value in artificial neural networks before converting it into SNN [Neil et al., 2016]. In fact, direct regularization of spike activity actually conflicts with the optimization goal, leading to a trade-off in performance and sparsity, as these methods are either limited to shallow networks or fail to effectively train highly sparse models [Narduzzi et al., 2022, Sakemi et al., 2023].

While pruning techniques are essential for training efficient SNNs, they cannot be separated from the underlying learning process. Specialized optimization techniques are required to ensure both efficient learning of the target function and eliminating as much redundant structure as possible. Most researches that focus on the pruning of SNNs adopts the supervise-learning-based approaches [Bohte et al., 2000], or more specifically, the RNN-like backpropagation-through-time (BPTT) algorithm [Wu et al., 2018]. In this method, the surrogate gradient approximation is proposed to smooth the non-differentiable firing function in the neuron [Lee et al., 2016, Shrestha and Orchard, 2018, Fang et al., 2021a, Neftci et al., 2019, Zenke and Vogels, 2021, Stewart and Neftci, 2022]. Some researches also adopted the event-driven backpropagation that is able to maintain the sparsity of the gradient [Fang et al., 2021b, Kim and Panda, 2020, Zheng et al., 2021, Lee et al., 2020, Deng et al., 2022, Mostafa, 2017, Bohte et al., 2000, Zhu et al., 2022, Zhang et al., 2021]. Another commonly used SNN training methods are ANN-SNN conversion approaches [Cao et al., 2015], which map the weights of a pre-trained ANN into an rate-coding SNN [Diehl et al., 2015, Sengupta et al., 2019, Han et al., 2020, Deng and Gu, 2021, Ding et al., 2021, Bu et al., 2022, Meng et al., 2022]. ANN-SNN conversion is the most practical training method to train SNNs on large-scale datasets since converted SNNs always have outstanding performance [Kim et al., 2020, Hao et al., 2023b,a].

### 2.2 Neuron model

In this paper, we use the widely adopted Leaky-Integrate-and-Fire model for SNNs [Izhikevich, 2004, Gerstner et al., 2014]. The discretized dynamics of membrane potential can be summarized by

$$x_i[t] = w_{ij}s_j[t] + b_i, \tag{1}$$
$$m_i[t] = \tau u_i[t-1] + x_i[t], \tag{2}$$
$$s_i[t] = H(m_i[t] - \theta_i), \tag{3}$$
$$u_i[t] = (1 - H(m_i[t] - \theta_i)) \cdot m_i[t], \tag{4}$$

where $j$ and $i$ indicate the two adjacent neurons and $w_{ij}$ and $b_i$ represent the weight and bias of the connection from neuron $j$ to neuron $i$. At arbitrary time-step $t$, the current input of neuron $i$ is $x_i[t]$ (Eq. 1). $m_i[t]$ and $u_i[t]$ represent the value of the membrane potential before and after neuronal firing. $H(\cdot)$ is the Heaviside step function. The neuron will emit a spike, denoting by the 0-1 scaler $s_i[t]$ when the membrane potential $m_i[t]$ reaches the firing threshold $\theta_i$. $\tau$ is the leaky parameter and if we explicitly set $\tau$ to 1, the neuron will degenerate to the non-leaky IF model. We demonstrate the hard-reset function in Eq. 4 that reduce the neuron membrane potential to the resting potential 0.

## 2.3 Spike activity pruning in SNNs

A general method for learning sparse activated models with activation regularization involves modifying the loss function with a penalty term that promotes sparsity in the network's activations. There are various techniques to implement such regularization over neuron activation, including applying L1/L2 regularization over the average spike rate per neuron [Neil et al., 2016, Yan et al., 2022, Deng et al., 2021], or using other regularizers [Narduzzi et al., 2022]. With the constraint of minimizing the spike activity, the general loss function of such activity-based pruning methods consists of both learning objectives for the specific task and the regularization term over the spike activity, which is

$$\mathcal{L}_{total} = \frac{1}{M} \sum_{k=1}^{M} \left( \mathcal{L}_{task} \left( \boldsymbol{f}(\boldsymbol{x}_k), \boldsymbol{y}_k \right) \right) + \lambda_s \sum_{i=1}^{N} \| \boldsymbol{s}_i(\boldsymbol{x}_k) \|_p \right). \tag{5}$$

We use $\mathcal{L}_{task}$ (term in the left) and $\mathcal{L}_{reg}$ (term in the right) denotes the task-specific loss and activation regularization loss, respectively. $M$ represents the number of training samples and $N$ are total neuron numbers. $\boldsymbol{x}_k, \boldsymbol{y}_k$ denotes the $k$-th data-label sample from the training dataset. $\boldsymbol{f}(\cdot)$ is the function that represents the input-output mapping of the model. The regularization term is $\lambda_s \sum_{i=1}^{N} \| \boldsymbol{s}_i(\boldsymbol{x}_k) \|_p$ is the summation of the $p$-norm of the current output spike vector with the penalty coefficient parameter $\lambda_s$, where vector $\boldsymbol{s}_i(\boldsymbol{x}_k)$ represents the output spike pattern of neuron $i$ across all time-steps when the model receives $\boldsymbol{x}_k$ as input. Many existing activity-based pruning methods for spiking neural networks can be considered as specific instances of this general framework, as they apply similar principles to add regularization based on their activity levels. However, using only the activation regularization technique cannot achieve a highly sparse model. While activation regularization can directly promote sparsity by adding constraints on parameters, this approach often leads to a trade-off between efficiency and accuracy. As the penalty term increases or the model becomes sparser, performance degradation becomes more severe, limiting its practical effectiveness.

## 3 Method

### 3.1 Learning dilemma of better performance and sparser spikes

The trade-off between the model performance and activation sparsity in artificial neural networks has been widely recognized as a challenge [Li et al., 2023, Georgiadis, 2019, Kurtz et al., 2020]. In SNNs, this issue is even more severe due to the sparser activations and inaccurate gradient estimation when using a surrogate gradient. Here we summarize two key challenges that significantly hinder the learning process in the joint optimization of task-specific loss and regularization loss. These two issues impede the effective optimization, limiting both the efficiency and performance of the model.

**Conflicts on optimization target** The first issue that hinders the learning process is the dilemma between learning more sparse spike activations and learning non-zero representations from the input data, as the optimization objectives of $\mathcal{L}_{task}$ and $\mathcal{L}_{reg}$ are inherently contradictory. Let us assume that in the $i$-th neuron, when the model receives the $k$-th data sample $\boldsymbol{x}_k$ from the training dataset, $\hat{\boldsymbol{y}}_i$ is the optimal output feature representation for neuron $i$ that we attempt to learn. The regularization over the spike patterns are L2 norm. Under such assumption, the learning objective can be rewritten as

$$\mathcal{L}_{total} = \frac{1}{M} \sum_{k=1}^{M} \left( \sum_{i=1}^{N} \| \boldsymbol{s}_i(\boldsymbol{x}_k) - \hat{\boldsymbol{y}}_i \|_2^2 + \lambda_s \sum_{i=1}^{N} \| \boldsymbol{s}_i(\boldsymbol{x}_k) \|_2^2 \right) \tag{6}$$

$$= \frac{1}{M} \sum_{k=1}^{M} \sum_{i=1}^{N} \left( \| \boldsymbol{s}_i(\boldsymbol{x}_k) - \hat{\boldsymbol{y}}_i \|_2^2 + \lambda_s \| \boldsymbol{s}_i(\boldsymbol{x}_k) - \boldsymbol{0} \|_2^2 \right). \tag{7}$$

The Eq. 7 illustrates a fundamental conflict between the target loss and the regularization term in the learning process. One the one hand, minimizing the target loss typically encourages the model to capture intricate patterns in the data and learning non-zero feature representations. The spike-activity regularization, on the other hand, encourages sparser spike patterns and less spike output. Therefore, there is an obvious trade-off between effectively learning an inner representation from input data and reducing spike activity at each layer. The reduction of the spike disrupts the representation learning at each layer since the regularization term inhibits spike firing. The trade-off between the

two objectives makes it challenging to simultaneously optimize for both effective representation learning and minimal spike activity.

**Gradient vanishing problem with sparse activity** The other issue that impedes the learning process of sparse SNNs comes from the learning process when using gradient-based learning. It has been observed that the gradient vanishing problem may significantly degrade the performance of the SNN when using Backpropagation-Through-Time (BPTT) [Fang et al., 2021a]. Furthermore, this issue becomes even more severe and prevalent when activity regularization is applied, as the sparsity of spike activity will further induce gradient vanishing and prevent the model from convergence. To illustrate this issue, we first provide a theoretical analysis showing that excessive regularization of the average firing rate can cause the training process to be trapped at a saddle point.

**Theorem 1.** *We define* $\boldsymbol{S} = \{\boldsymbol{w}|\forall l, t \ \ \boldsymbol{s}^l[t] = \boldsymbol{0}\}$ *as the set of all parameter sets such that the output spike at arbitrary layers and time-steps is all zero, then the* $\boldsymbol{S}$ *is a set of saddle points for the optimization of the total loss* $\mathcal{L}_{total}$, *which is*

$$\forall i \ \ \sum_t \boldsymbol{s}_i[t] \to 0 \ \ \Rightarrow \ \ |\frac{\partial \mathcal{L}_{total}}{\partial w_{ij}}| \to 0, \tag{8}$$

$$\frac{\partial \mathcal{L}_{total}}{\partial w_{ij}} = 0, \ \ \forall w_{ij} \in \boldsymbol{S} \tag{9}$$

$\boldsymbol{S}$ denotes the set of all parameter sets such that the output spike at an arbitrary layer and time is all zero. As described in the Eq. 9, we can prove that, in practice, the $\boldsymbol{S}$ is a set of critical points for the optimization of the total loss $\mathcal{L}_{total}$, since the gradient of the weight vectors with respect to the total loss $\nabla_{\boldsymbol{w}} \mathcal{L}_{total}$ is all zero and normally it is not the global minima of the total loss. Therefore, any $\boldsymbol{w} \in \boldsymbol{S}$ is a saddle point of the optimization problem, and the gradient will converge to zero when the spike output is zero. Detailed proof will be provided in the supplementary material.

This theorem identifies a set of saddle points in the loss function, highlighting a critical issue in the optimization process. When regularizing spike activity, the model might easily fall into the local minima with extremely sparse activity. In this case, the learning process will stop since the gradient comes to zero, and the optimization process will be trapped at the saddle point. This phenomenon emphasizes the need for carefully designed regularization strategies to avoid gradient vanishing and maintain effective learning dynamics.

## 3.2 Adaptive threshold LIF neuron

In the previous section, we identify two key issues that hinder the training of the sparse activated spiking neural networks from two perspectives: the conflict in optimization target and the potential gradient vanishing problem during optimization. Therefore, in this section, to alleviate the impact of the learning dilemma, we provide a feasible solution by introducing the Adaptive Threshold Neuron (AT-LIF), which includes: 1) introducing the adaptation mechanism to the neuron threshold; 2) using current-based output with 0-$\theta$ value instead of 0-1 spike output.

With the adaptation mechanism, the threshold value $\theta_i$ evolves throughout training and adapt to the spike frequency at each iteration. The threshold thus becomes a learnable parameter during the training process and plays a role of both controlling the firing frequency of the neurons and the efficacy of transmitting outputs to the next layer:

$$o_i[t] = \theta_i \cdot H(m_i[t] - \theta_i) = \theta_i s_i[t], \tag{10}$$
$$x_i[t] = w_{ij} o_j[t] + b_i. \tag{11}$$

We use $o_i[t] \in \{0, \theta_i\}$ to represent the output of the AT-LIF neuron instead of the $s_i[t]$ in the common LIF neuron (Eq. 1, 3), which scales up the neuron output while the average activity is low. Note that this mechanism does not imply graded output spikes, as the scaling by the threshold value does not alter the binary nature of the spike outputs. During inference, when the thresholds are fixed, the scaling can be absorbed into the subsequent layer's weights, ensuring full compatibility with all types of neuromorphic hardware. Intuitively, increasing the threshold reduces the firing frequency while preserving the average output value. Before introducing the detailed adaptive algorithm, we first demonstrate how threshold adaptation gradually reduces the neuron activity (Fig. 2). In this toy example, we continually add constant input to the AT-LIF neuron and vanilla LIF neuron and compare

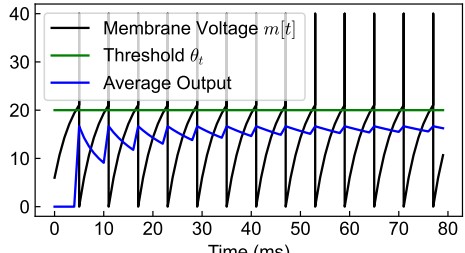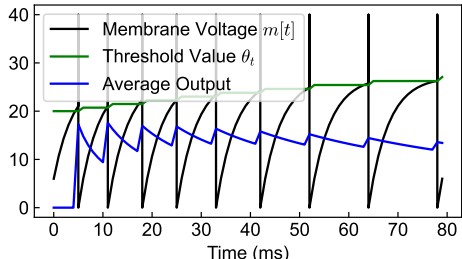

Figure 2: Left: Behavior of the vanilla LIF neuron; Right: Behavior of the AT-LIF neuron. In both figures, the black curve represents the change of the membrane potential, while the green curve represents the threshold value and the blue curve represents the average neuronal output over time.

both the output and average firing rate of these two different types of neurons. In the AT-LIF model, the threshold will increase slightly after each output spike, while the threshold of the vanilla LIF will remain constant. Such adaptation of the threshold helps reduce the firing rate while preventing a drop of the average output of the neuron. These results demonstrate the potential of the AT-LIF model to suppress output spiking activity while preserving stable output dynamics and faithful neuronal representation of input information.

### 3.3 Threshold adaptation for sparse activity

In practice, we incorporate the threshold adaptation mechanism into the Backpropagation-Through-Time (BPTT) algorithm [Wu et al., 2018] by treating the threshold at each layer as a learnable parameter during training. In order to constrain the spike activity, we regularize the threshold value to minimize the neuron firing rate. To constrain spike activity, we regularize the threshold values by minimizing the firing rate at each layer. Specifically, the thresholds are optimized to minimize both the task-specific loss and the activity regularization loss ($\arg\min_{\boldsymbol{\theta}} \mathcal{L}_{total}$), while the synaptic weights are updated solely to minimize the task-specific target loss. Based on this formulation, we derive the threshold update rule at each update iteration in Eq. 12.

$$\Delta\theta_i = \eta \left( \frac{\partial \mathcal{L}_{task}}{\partial \theta_i} + \lambda_\theta \sum_{t=1}^{T} s_i[t] \frac{\partial o_i[t]}{\partial \theta_i} \right), \tag{12}$$

where $T$ is the inference time-steps for the SNN model and $\eta$ is the learning rate during the training process. $\frac{\partial s_i[t]}{\partial \theta_i}$ can be estimated by the surrogate function. To distinguish from the regularization-based method over average spike activity, we use $\lambda_\theta$ to represent the coefficient parameter of the threshold adaptation. In practice, we consider each $\theta_i$ as a learnable parameter that is involved in the training process, while also manually regularize its value at each iteration according to Eq. 12. We also demonstrate the forward/backward information flow for AT-LIF neuron in right part of the Fig 3. The full derivation and pseudo-code of the overall algorithm are provided in the supplementary material.

As for the surrogate function, we consider the triangle-shaped function as surrogate of the Heaviside function [Esser et al., 2016] to estimate the derivative of the firing function of the AT-LIF neuron (Eq. 13). To prevent gradient explosion, the upper bound of the surrogate function is set to 1 while the design principle of the triangle-shaped surrogate function is strictly followed. Leveraging this surrogate, along with the proposed learning framework and adaptation rules for AT-LIF, we are able to train high-performance spiking neural networks with sparser activation.

$$\frac{\partial o_i[t]}{\partial m_i[t]} = -\frac{\partial o_i[t]}{\partial \theta_i} = \max \left( 1 - \frac{|m_i[t] - \theta_i|}{\theta_i}, 0 \right). \tag{13}$$

### 3.4 Stabilizing training with AT-LIF neuron

In this section, we demonstrate how the AT-LIF neuron addresses two key challenges that limit the performance of sparse SNN models. Specifically, the AT-LIF neuron resolves the inherent conflict

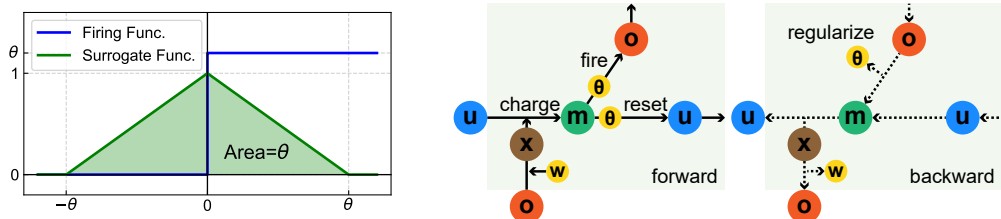

Figure 3: Left: The surrogate function for AT-LIF neuron. The blue curve represents the firing function while the green curve represents the surrogate derivative. Right: Illustration of the forward pass and backward pass of the AT-LIF neuron using the Back-Propagation-Through-Time algorithm.

between competing learning objectives and mitigates the gradient vanishing problem encountered when training highly sparse networks.

**Solving the conflict of optimization target**
As discussed in the above section, training sparse spiking neural networks typically involves jointly optimizing the task objective and the sparsity constraint as $\boldsymbol{\theta}, \boldsymbol{w} = \arg\min_{\boldsymbol{\theta}, \boldsymbol{w}} \mathcal{L}_{total}$, which leads to substantial conflict on optimization target. In our method, however, we introduce the AT-LIF neuron, which outputs a scaled binary signal rather than a standard binary spike.

This scaling mechanism helps maintain a steady average neuron output while simultaneously reducing the firing rate (as illustrated in Fig 2). This property allows us to decouple the learning process while also maintaining stable learning process. Specifically, instead of jointly optimize the $\mathcal{L}_{total}$, we separate the optimization by learning the weight parameters based on the task loss as $\boldsymbol{w} = \arg\min_{\boldsymbol{w}} \mathcal{L}_{task}$ while simultaneously regularize the threshold to reduce spike activity as $\boldsymbol{\theta} = \arg\min_{\boldsymbol{\theta}} \mathcal{L}_{total}$.

The conflict does not necessarily arise for learning of weight parameters as the model can still effectively learn from the output representation while we simultaneously maximize the threshold to promote sparser firing events. Consequently, the inherent conflict between the optimization objectives can, in principle, be resolved when adaptive threshold neurons are employed. As a result, the network can effective learning representations while reducing spike counts during the optimization process.

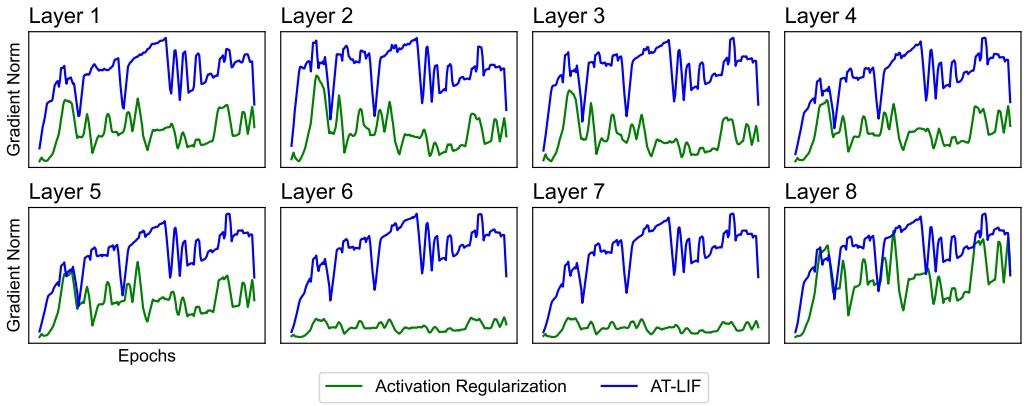

Figure 4: The 2-norm of the gradient value in the first 8 Convolutional layer.

**Alleviating the gradient vanishing problem**
On the other hand, the gradient vanishing problem will be alleviated since the gradient will no longer necessarily approach zero as the threshold value increases. One possible theoretical explanation is that the saddle point assumption (Theorem 1) no longer holds true as we continually increase the threshold rather than directly constrain the spike activity. We further empirically demonstrate this phenomenon by comparing the change of the gradient value with respect to the training iterations in different activity pruning methods before and after the adoption of the AT-LIF neuron. We trained multiple SNN models using AT-LIF neuron with threshold adaptation and vanilla LIF neuron with activation regularization on CIFAR-10 dataset with ResNet-20 structure and selected two models

with similar average firing rates after training for fair comparison. Here the model using AT-LIF achieved 91.06% accuracy and 0.043 average firing rate, while the model using vanilla LIF only achieved 88.31% accuracy and a larger average firing rate of 0.044. We demonstrate the comparison of the gradient values of the two models at each training iteration (Fig. 4). The green curve represents the normalized gradient 2-norm value of the weight vector in the corresponding convolution layer in the model with vanilla LIF and activation regularization, while the blue curve represents that of the model adopting AT-LIF neuron. Here the blue curve is always above the green curve and in most layers, the gradient value of the model using activation regularization is much closer to zero while the gradient value in the model with AT-LIF neuron still maintains consistency across layers. This indicates that using AT-LIF can effectively mitigate the gradient vanishing problem. Moreover, this is also confirmed by the fact that the final training results of the AT-LIF model achieve better performance while achieving similar sparsity.

## 4 Experiments

### 4.1 General experimental setting

We provide the general hyper-parameters and the neuron parameter setting in here and the supplementary material for better reproducibility. Following the previous study [Chen et al., 2022, Shi et al., 2024], we select a similar architecture for evaluation, including 6 Conv, 2FC (CIFAR-10), SEW-ResNet18(ImageNet, ImageNet-100), and VGGSNN(DVS-CIFAR10). We also utilize the standard ResNet-20 for the CIFAR dataset [He et al., 2016].

To demonstrate the effectiveness of the pruning method, we will use the following metrics in the below table, including classification accuracy (Acc.), average firing rate (Avg. FR), total parameters (Params) and total synaptic operation numbers (SOPs). The definition for the average firing rate $r = \sum_{k=1}^{M} \sum_{i=1}^{N} \sum_{t=1}^{T} s_i(\boldsymbol{x}_k)[t]$, where $x_k$ are data samples from the test set and $s_i(\boldsymbol{x}_k)[t]$ is the spike value in each neuron at each time-step with given input $\boldsymbol{x}_k$. The synaptic operation (SOP) quantifies the total number of computational events in spiking neural networks, corresponding to the number of times a spike is transmitted across a synapse [Hu et al., 2018, Fang et al., 2021a]. Additionally, we report the number of parameters left after applying weight-based pruning.

### 4.2 Compare with existing methods

| Dataset | Arch. | Method | T | $\lambda_s$ | $\lambda_\theta$ | Acc.(%) | Avg FR. | SOPs(M) |
|---------|-------|--------|---|-------------|------------------|---------|---------|---------|
| | ResNet-20 | Vanilla | 8 | 0 | 0 | 92.89 | 0.236 | 256.69 |
| | ResNet-20 | AR | 8 | 0.01 | 0 | 91.31 | 0.060 | 57.26 |
| | ResNet-20 | AR | 8 | 0.02 | 0 | 88.31 | 0.044 | 41.86 |
| CIFAR-10 | ResNet-20 | AT-LIF | 8 | 0 | 1e-3 | 92.05 | 0.053 | 61.70 |
| | ResNet-20 | AT-LIF | 8 | 0 | 2e-3 | 91.06 | 0.043 | 48.85 |
| | ResNet-20 | AT-LIF | 8 | 0 | 5e-3 | 88.31 | 0.033 | 39.33 |
| | ResNet-20 | AT-LIF | 8 | 0.01 | 1e-3 | 90.70 | 0.025 | 29.23 |
| | SEW-ResNet18 | Vanilla | 4 | 0 | 0 | 83.44 | 0.138 | 1276.42 |
| | SEW-ResNet18 | AR | 4 | 0.01 | 0 | 82.94 | 0.083 | 840.90 |
| ImageNet-100 | SEW-ResNet18 | AR | 4 | 0.05 | 0 | 82.26 | 0.043 | 469.00 |
| | SEW-ResNet18 | AT-LIF | 4 | 0.01 | 1e-4 | 82.40 | 0.038 | 463.65 |
| | SEW-ResNet18 | AT-LIF | 4 | 0.02 | 2e-4 | 81.12 | 0.025 | 328.34 |
| | VGGSNN | Vanilla | 10 | 0 | 0 | 82.70 | 0.080 | 1066.40 |
| DVS-CIFAR10 | VGGSNN | AR | 10 | 0.005 | 0 | 81.30 | 0.055 | 673.06 |
| | VGGSNN | AT-LIF | 10 | 0 | 5e-4 | 82.20 | 0.043 | 626.57 |
| | VGGSNN | AT-LIF | 10 | 0.001 | 5e-4 | 82.10 | 0.039 | 587.68 |

Table 1: Comparison of AT-LIF and direct activation-regularization-based methods on CIFAR-10, ImageNet-100, and DVS-CIFAR10 datasets.

In this section, we compare our proposed model with other existing SNN pruning approaches to evaluate its performance and highlight its advantages. Here we will present two tables for fair

comparison. The first table compares our pruning method with existing activity regularization approaches, all of which focus only on reducing spike activity or average firing rate. The second table reports results under the setting of unconstrained structural pruning, where activity-based pruning is combined with connection-level or neuron-level pruning methods. Specifically, we integrate our method with the weight pruning approach STDS [Chen et al., 2022] to demonstrate its potential to learn highly efficient SNNs.

**Spike activity pruning**
Since existing activity regularization methods primarily focus on penalizing spike activity, we adopt L1 spike regularization as a representative regularization-based approach for comparison. Tab. 1 presents a comprehensive evaluation of our proposed AT-LIF model (AT-LIF) against two baselines: the vanilla model (Vanilla) and vanilla model trained with spike activity regularization (AR), across three benchmark datasets: CIFAR-10, ImageNet-100, and DVS-CIFAR10. The columns "$\lambda_s$" and "$\lambda_\theta$" are the regularization coefficients for the spike activity term (Eq. 7) and the coefficient for the threshold adaptation mechanism, respectively. The column $T$ refers to the simulation timestep for SNN.

Across all tasks, the AT-LIF neuron demonstrates a strong capability to reduce spike activity and computational cost while maintaining high classification accuracy compared with regularization-based methods. On CIFAR-10, AT-LIF achieves a similar accuracy to the unpruned Vanilla model, while reducing the average firing rate by approximately 89.4% and lowering SOPs by nearly 88.6%. Compared to activation regularization, AT-LIF achieves lower SOPs and firing rates while maintaining higher accuracy. Similar trends are observed on the ImageNet-100 dataset, where AT-LIF maintains competitive accuracy with 1% accuracy loss while significantly reducing SOPs and average firing rate. If sacrificing 2.3% more accuracy is acceptable, the reduction of average firing rate on the AT-LIF model will be more than 5.5 times. On the event-based DVS-CIFAR10 dataset, AT-LIF again achieves efficient spike suppression with substantial reductions in SOPs (up to 45% fewer than Vanilla) and firing rates, while preserving accuracy. These results collectively highlight the effectiveness of AT-LIF in achieving sparse, energy-efficient SNNs without sacrificing much performance, outperforming conventional activity regularization techniques.

| Dataset | Arch. | Pruning | Method | Acc.(%) | Avg FR. | Param(M) | SOPs(M) |
|---|---|---|---|---|---|---|---|
| CIFAR-10 | 6 Conv, 2 FC | Conn | STDS | 90.71 | 0.072 | 0.12 | 45.48 |
| | | | | 89.40 | 0.059 | 0.07 | 28.38 |
| | 6 Conv, 2 FC | Neu;Conn | Unstru. | 91.74 | 0.051 | 9.48 | 15.15 |
| | | | | 90.76 | 0.049 | 6.93 | 8.49 |
| | 6 Conv, 2 FC | Act;Conn | AT-LIF | 90.72 | 0.024 | 0.19 | 30.11 |
| | | | | 90.20 | 0.020 | 0.09 | 16.88 |
| ImageNet | SEW-ResNet18 | Conn | STDS | 58.90 | 0.189 | 1.84 | 461.26 |
| | SEW-ResNet18 | Neu;Conn | Unstru. | 59.23 | 0.106 | 4.38 | 271.98 |
| | SEW-ResNet18 | Act;Conn | AT-LIF | 59.61 | 0.063 | 2.22 | 310.53 |

Table 2: Comparison of the composite AT-LIF method with existing SNN pruning frameworks.

**Pruning both activity and weights**
We also integrate our method with the STDS [Chen et al., 2022] to achieve the pruning of both weights and neuron activity. STDS operates by reparameterizing the weights and progressively eliminating those with small absolute values during training, leading to a sparse connectivity pattern. Our integration is straightforward: we replace the standard LIF neurons with AT-LIF neurons throughout the network, and apply threshold adaptation as described in our method. Meanwhile, the STDS pruning algorithm is used to train the synaptic weights and induce weight sparsity. This combination allows us to simultaneously achieve both activity sparsity and weight sparsity. We compare the composite approach with reproduced implementations of existing SNN pruning frameworks, including unstructured weight pruning (STDS) and unstructured weight and neuron pruning (Unstru.) [Shi et al., 2024]. We compare the above methods in accuracy, average firing rate, parameter numbers and total SOPs. The final results are demonstrated in Tab. 2.

On CIFAR-10, the composite AT-LIF method achieves comparable accuracy to both STDS and Unstru., while significantly reducing the average firing rate. Notably, AT-LIF achieves the lowest average firing rate at 0.020 and reduces SOPs to 16.88M, representing a substantial gain in computational

efficiency. On ImageNet, AT-LIF again achieves the lowest firing rate (0.063) and a favorable balance between accuracy and efficiency, with SOPs reduced compared to STDS (461.26M) and close to those achieved by the unstructured method (271.98M). Overall, the results highlight the strength of AT-LIF in achieving efficient sparse SNNs with low energy cost, while maintaining performance across datasets and architectures. Compared with the Unstru. that aims to reduce SOPs and STDS that aims to reduce parameters, although our method is not optimal in every individual metric, it strikes an effective balance between spike sparsity and weight sparsity, achieving the best performance across these two key indicators. This highlights the flexibility and effectiveness of the proposed AT-LIF model in enabling both activity and weight sparsity when combined with weight pruning techniques.

## 4.3 Ablation study

An ablation study was conducted to evaluate the impact of specific components of the AT-LIF neuron model on the overall performance of the SNNs. We systematically remove or alter the two components as mentioned in Sec. 3.2 and therefore evaluate their individual contributions. The Tab. 3 demonstrates

| CurO. | Adap. | $\lambda_\theta$ | Acc.(%) | Avg. FR | SOPs(M) |
|---|---|---|---|---|---|
| $\times$ | $\times$ | 0.0 | 92.89 | 0.236 | 256.69 |
| $\checkmark$ | $\times$ | 0.0 | 93.32 | 0.237 | 258.95 |
| $\times$ | $\checkmark$ | 1e-3 | 50.99 | 0.110 | 126.13 |
| $\checkmark$ | $\checkmark$ | 1e-3 | 92.05 | 0.054 | 61.70 |

Table 3: Ablation study for AT-LIF neuron on CIFAR-10 dataset.

the model performance and efficiency under different neuron configurations. Here column "CurO." represents whether the neuron has 0-1 output or 0-$\theta$ output while column "Adap." indicates whether the threshold adaptation mechanism is enabled. The model without the current-based output feature fails to converge when combined with the adaptation mechanism, indicating that the resulting sparse spike output hinders effective model learning. Models that applied the adaptive threshold exhibit a significant efficiency advantage over the one with the vanilla LIF neuron, with a notable reduction in overall computational cost. The model with AT-LIF neuron achieves the most outstanding results across all experiments as it is four times more efficient compared with the baseline model in terms of spike sparsity and energy cost. Results show that both current-based output and threshold adaptation are crucial for maintaining an optimal trade-off between minimizing excessive spike activity and preserving the model's ability to learn effectively.

## 5 Conclusion and limitation

In this paper, we propose a novel technique to significantly reduce spike activity in SNNs while preserving their performance. By effectively suppressing spike events, our method substantially promotes the sparsity of spike outputs and decreases required synaptic operations. As the proposed method focuses on reducing spike activity, it is also compatible with weight pruning or other model compression techniques. The proposed method potentially enables efficient mapping onto neuromorphic hardware or even on modern GPU architectures.
**Limitation** While this work introduces the concept of exploiting the sparse spike activity of SNNs, it still remains at the algorithmic level. The next step may involve exploring practical hardware implementations and evaluating the effectiveness of the approach in real-world applications.

## 6 Acknowledgement

This work is funded by National Natural Science Foundation of China (62422601, U24B20140, and 62088102), Beijing Municipal Science and Technology Program (Z241100004224004) and Beijing Nova Program (20230484362, 20240484703), and National Key Laboratory for Multimedia Information Processing.

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
