# OpenReview forum: "Activity Pruning for Efficient Spiking Neural Networks"
_NeurIPS.cc/2025/Conference — NeurIPS 2025 poster_

### Official Review · Reviewer_7Qnw · 2025-06-24

**Clarity:** 3
**Significance:** 3
**Originality:** 2
**Rating:** 4
**Confidence:** 3

**Summary:**

This paper designs AT-LIF, a novel activity pruning method for SNN. The paper first discusses the two major challenges in activity regularization pruning method and then addresses these issues by introducing (1) learnable threshold for the neuron, and (2) current-based spike output. Evaluations on three datasets demonstrate the effectiveness of proposed methods.

**Questions:**

The following points could be addressed to further improve the clarity and impact of the work.

1.The inclusion of more SOTA pruning methods is encouraged.
2.In Section 3.4, the authors state that the model can simultaneously learn from the output representation and promote spike sparsity, thereby resolving the optimization conflict. The authors could provide a detailed explanation on why this trade-off is achievable.
3.The proposed AT-LIF replaces binary output with a current-based output. Could the authors clarify how this design choice how this design contributes to addressing each of the two main challenges discussed in the Sec 3.1?
4.The choice of hyperparameter \( \lambda_s \) for both the AR baseline and AT-LIF appears somewhat arbitrary and varies within and across datasets without clear explanation. Could the authors clarify how these values were selected?

**Ethical Concerns:**

["NO or VERY MINOR ethics concerns only"]

**Final Justification:**

Authors have resolved my concerns in the rebuttal stage.

**Limitations:**

yes

**Paper Formatting Concerns:**

I did not notice formatting issues.

**Quality:**

2

**Strengths And Weaknesses:**

Strength:
1. Clear discussion of challenges. The paper presents a well-motivated discussion of the existing challenges in training SNN. It analyzes conflict between minimizing spike activity and preserving task-specific performance and further identifies the gradient vanishing problem. These challenges are theoretically analyzed, providing a strong motivation for the proposed AT-LIF.
2. The proposed method is simple and easy to implement. It addresses the identified challenges by modifying only the neuron's firing threshold and output range, without introducing complex architectural changes. This lightweight design makes the method promising for practical applications.


Weaknesses:
1.Insufficient comparative experiments. The paper only compares the proposed AT-LIF with a basic activation regularization baseline and two weight pruning methods. This narrow comparison makes it difficult to assess the effectiveness.
2.Unclear explanation about conflict resolution. In Section 3.4, the authors state that “the conflict between the two terms does not necessarily arise, as the model can still effectively learn from the output representation while we simultaneously maximize the threshold to promote sparser firing events” (Line225-227). However, the paper does not explain why or how the model can simultaneously achieve effective representation learning and sparse spike activity. As a result, the justification for "resolving the optimization conflict” is not convincing.
3.Minor clarity issues. Eq.10 and Fig.3 (Right) are not described or referenced in the main text.

---

> ### Author Rebuttal · Authors · 2025-07-29
>
> # Response to Reviewer 7Qnw
> > Insufficient comparative experiments. The inclusion of more SOTA pruning methods is encouraged.
>
> We sincerely thank the reviewer for this constructive suggestion. In the revised manuscript, we will incorporate additional comparisons with more recent and representative pruning techniques for both activity pruning and combined activity/weight pruning.
>
> For activity pruning, as discussed in the manuscript, most prior approaches rely on direct regularization over spike activity. To provide a more comprehensive evaluation, here we include comparisons with additional baseline techniques, including: (1) L1 regularization over average firing rate (adopted by [1] as part of its approaches), and (2) Hoyer regularization over output spike patterns [2]. These methods are evaluated using ResNet-20 on the CIFAR-10 dataset (Table below). Notably, our method achieve the superior sparsity compared to all existing results.
>
> |          | Accuracy | SOPs (M) | Avg. FR. |
> | -------- | -------- | -------- | -------- |
> | L1 Reg [1]    | 91.27    | 60.21    |  0.062   |
> | Hoyer Reg [2] | 88.62    | 76.79    |  0.081   |
> | AR [Baseline] | 91.31    | 57.26    |  0.060   |
> | AT-LIF [Ours] | 91.06    | 48.85    |  0.043   |
>
> For combined pruning (i.e., both weights and spike activity), we also include three additional state-of-the-art baselines on CIFAR-10 for comparison: ADMM[1], GradR [3], and ESL-SNN[4].
> |          | Arch.    |   T     | Accuracy | SOPs (M) | Param (M)|
> | -------- | -------- |-------- | -------- | -------- | -------- |
> | ADMM [1]   | 7 Conv, 2 FC | 8 | 90.19    | 107.97   |  15.54   |
> | Grad-R [3] | 6 Conv, 2 FC | 8 | 91.37    | 87.73    |   0.86   |
> | ELS-SNN [4]   | ResNet-19 | 8 | 90.90    | 108.89   |   0.63   |
> | AT-LIF [Ours] | 6 Conv, 2 FC | 8 | 90.72 | 30.11    |   0.19   |
>
> We demonstrate that AT-LIF, when integrated with weight pruning techniques, consistently achieves a favorable balance between accuracy and spike sparsity. We will incorporate all these expanded results and discussions in the revised manuscript to better support the empirical claims.
>
> > Unclear explanation about conflict resolution. The authors state that the model can simultaneously learn from the output representation and promote spike sparsity. The authors could provide a detailed explanation on why this trade-off is achievable.
>
> We thank the reviewer for this thoughtful comment, and we are happy to provide further clarification.
>
> As discussed in the manuscript, training sparse spiking neural networks typically involves jointly optimizing the task objective and the sparsity constraint:
> $$\theta,w = \arg\min_{\theta, w} \mathcal L_{total} ,~~\text{where}~ \mathcal L_{total} = \mathcal L_{task} + \mathcal L_{sparsity}$$
> In our method, however, we introduce the AT-LIF neuron, which outputs a scaled binary signal rather than a standard binary spike. This scaling mechanism helps maintain a steady average neuron output while simultaneously reducing the firing rate(as illustrated in Figure 2 of the manuscript). This property allows us to decouple the learning process. Specifically, since most deep spiking neural networks rely on rate coding, we assume that regularizing the threshold does not significantly affect the information encoded within the layers. Therefore, instead of jointly optimize the $\mathcal L_{total}$, we seperate the optimization by learning the weight parameters based on the task loss while simotenously regularize the threshold to reduce spike activity.
>
> $$w=\arg\min_{w} \mathcal L_{task}; \quad \theta=\arg\min_{\theta} \mathcal L_{sparsity}.$$
>
> As a result, the network can effective learning representations while reducing spike counts during the optimization process. We will revise Section 3.4 to better highlight this mechanism and clarify how it enables a more graceful trade-off between accuracy and sparsity.
>
> > Could the authors clarify how "current-based output" design contributes to addressing each of the two main challenges?
>
> Thank you for your insightful comments. As discussed above, the AT-LIF model enables current-based output, defined as $o_j[t] = s_j[t] \cdot \theta_j$.  During training, we assume that the average current-based output $\sum_{t=1}^{T}  o_j[t]/T$ remains relatively stable despite the threshold regularization. This stability helps reduce the conflict between representation learning and sparsity penalty.
>
> Moreover, since the neuron’s output magnitude remains steady throughout training, this also helps alleviate the gradient vanishing problem commonly seen in sparse spiking neural networks. Specifically, in the AT-LIF model, the derivative with respect to the synaptic weight $w_{ij}$ can be bounded as follows
> $$\left |\frac{\partial \mathcal{L}}{\partial w_{ij}} \right | = \left |\sum_{t=1}^{T} \frac{\partial \mathcal{L}}{\partial m_{i}[t]} o_j[t] \right | \leqslant
> \left (\sum_{t=1}^{T}  o_j[t] \right) \left(\max_t \left | \frac{\partial \mathcal{L}}{\partial m_{i}[t]} \right | \right ).$$
> Since $\sum_{t=1}^{T}  o_j[t]$ is assumed to remain steady during training, the upper bound of the gradient does not approach zero. This suggests that gradient vanishing is not likely to occur, thereby enabling more stable and effective optimization.
>
> For further clarification, we now provide a more detailed analysis of how both the task loss and the sparsity loss evolve during training under different neuron output behavior (Table Below). We evaluate two ResNet-20 models with different neuron output behavior on CIFAR-10 dataset and display the change of task loss value, sparsity loss value and average gradient 2-norm in the following Table.
>
> |                  |  | Ep0  | Ep1  | Ep2 | Ep3 | Ep4 | ... | Ep10 |
> | ------------------------ |-- |--- | --- | --- | --- | --- |--- |--- |
> | w/o current-based |  Task Loss |1.77|1.47|1.37|1.43|1.56|...|2.30|
> | w/o current-based | Sparsity Loss |29.1|19.8|16.4|15.2|14.9|...|13.9|
> | w/o current-based |Gradient Norm |2.69|3.22|3.82|0.955|0.429|...|1.27e-4|
> | w/ current-based output  | Task Loss |1.79|1.51|1.32|1.21|1.12       |...|0.810|
> | w/ current-based output  | Sparsity Loss|29.3|20.1|16.5|14.5|12.9      |...|9.49|
> | w/ current-based output  | Gradient Norm |2.67|3.54|3.78|3.65|3.67 |...|12.3|
>
> From the results, we observe that without the current-based output, the model struggles to balance the learning objectives and suffers from a obvious gradient vanishing problem. Additionally, the ablation study presented in Section 4.2 provides more direct evidence of the importance of the current-based output.
>
> We will further expand the discussion on this point in the revised manuscript.
>
> > Minor clarity issues. Eq.10 and Fig.3 (Right) are not described or referenced in the main text.
>
> Equation 10 was intended to define the iterative update of the neuron threshold during training. It expresses how the threshold θ is incrementally adjusted at each training step as part of the threshold adaptation mechanism. To improve readability, we will remove Equation 10 and instead explain the threshold update more clearly through narrative description in the revised version.
>
> Figure 3 (Right) illustrates the forward and backward computational flow of the AT-LIF neuron during training using Backpropagation Through Time (BPTT). We will revise the manuscript to explicitly reference and discuss this figure in the main text to better guide the reader through the training dynamics of AT-LIF.
>
> > The choice of hyperparameter ( \lambda_s ) for both the AR baseline and AT-LIF appears somewhat arbitrary and varies within and across datasets without clear explanation. Could the authors clarify how these values were selected?
>
> We thank the reviewer for this valuable question. We acknowledge that our method introduces an additional coefficient $\lambda_t$, which controls the strength of the threshold regularization. In practice, for the AR baseline, $\lambda_s$ (the coefficient for activation regularization) was chosen based on commonly adopted values in previous works, and further adjusted via grid search to achieve a reasonable sparsity–accuracy trade-off. $\lambda_t$ controls the strength of the threshold adaptation, can be tuned via a simple grid search or empirically set based on prior experience.
>
> However, we would like to emphaize that unlike weight pruning, it is hard to directly control the desired sparsity of the spike activity, as different input sample triggers distinct pattern of spikes, resulting in different activation sparsity. Thus empirically setting the regularization coefficient for the penalty term is a more practical and feasible solution.
>
> Moreover, in our method, the convergence of the model is more robust to extreme values of the regularization coefficient. For example, with $\lambda_t=1e-3$, ResNet-20 on CIFAR-10 achieves 92.05% accuracy with 61.7M SOPs. Even when $\lambda_t$ is increased to $1e-1$, the model is still able to converge, achieving 45.52% accuracy with only 9.19M SOPs and 0.007 average firing rate. In contrast, for direct activation regularization, the model achieves 91.31% accuracy with 57.26M SOPs when $\lambda_s = 1e-2$, but fails to converge when $\lambda_s \geq 0.03$. This highlights the greater stability and flexibility of our method during hyper-parameter selection.
>
> We appreciate the reviewer’s feedback and will clarify our hyper-parameter selection strategy and its implications more explicitly in the revised manuscript.
>
> **Reference**
>
> [1] Deng, Lei, et al. "Comprehensive snn compression using admm optimization and activity regularization." TNNLS, 2021.
>
> [2] Narduzzi, Simon, et al. "Optimizing the consumption of spiking neural networks with activity regularization." ICASSP, 2022.
>
> [3] Chen, Yanqi, et al. "Pruning of Deep Spiking Neural Networks through Gradient Rewiring." IJCAI, 2021.
>
> [4] Shen, Jiangrong, et al. "ESL-SNNs: An evolutionary structure learning strategy for spiking neural networks." AAAI, 2023.

---

> > ### Comment · Reviewer_7Qnw · 2025-08-04
> > **authors have tried to resolved my concerns**
> >
> > thanks for your detail response to my concerns.

---

### Official Review · Reviewer_CMEG · 2025-06-27

**Clarity:** 2
**Significance:** 3
**Originality:** 3
**Rating:** 5
**Confidence:** 3

**Summary:**

The paper aims to improve efficiency of SNN learning while conserving effectiveness. Proposed AT-LIF algorithm reduces spiking activity using sparse regularizer and threshold adaptation.
The method is experimentally validated on several datasets widely used in deep learning research.

**Questions:**

Eq (6) why is the output of the i-th neuron, s_i, a vector? Also, y_hat should have both i and k indices.

There 1: Why is it derivative in (8) and gradient in (9)? Are (8,9) saying the same thing? Loose index j in (8).
Zero gradient is not exactly the same as a saddle point.
Maybe give a hint of the proof in the main text.

Line 179: subscript ()_t is never used

Line 196 BPTT requires memory of previous time states? Please clarify.

Doesn’t it make sense to do overall evaluation first, then ablation?

Line 260 what’s synaptic operation numbers?

Line 271: if the threshold \theta is constant, what’s the difference if the signal is 0-1 or 0-\theta?

Line 272: what’s synaptic output feature?

Line 281: what is SOTA?

Table 2: some of the best AT-LIF results are with \lambda_s=0, i.e. no sparsity. Please comment.
What is the column T?

Please give more details how your method is integrated with other methods in the experiments.

**Ethical Concerns:**

["NO or VERY MINOR ethics concerns only"]

**Final Justification:**

Authors response to my comments was detailed and convincing.

**Limitations:**

If the method is implemented on a SNN platform, that platform should allow for graded spikes.

**Paper Formatting Concerns:**

No major issues

**Quality:**

3

**Strengths And Weaknesses:**

Suggested algorithm is original and ingenious. The issue of vanishing gradient is duly addressed. Experiments demonstrate significant reduction of spiking activity with minimal loss of accuracy.
The design assumes graded spiking, which I think should be stated upfront.
The clarity is lacking, some terminology unclear, and overall leaves impression of hurried writing. This makes it difficult to evaluate the technical content properly.

---

> ### Author Rebuttal · Authors · 2025-07-29
>
> # Response to Reviewer CMEG
>
> > The clarity is lacking, some terminology unclear, and overall leaves impression of hurried writing.
>
> We appreciate the reviewer’s efforts to help improve the quality of our manuscript. We will carefully revise the entire manuscript to enhance clarity.
>
> > Clarify for the graded spike in AT-LIF.
>
> We sincerely thank the reviewer for pointing this out and apologize for the ambiguity in our original description. We would like to clarify that our method does not necessarily rely on graded spikes. While we replace the 0/1 spike output with a 0/θ value for stability and optimization purposes, the output remains binary in nature, since the neuron either fire or do not. Importantly, all neurons within the same layer share a single threshold value, which allows the scaled output to be removed during inference by absorbing the scaling factor into the synaptic weights of the subsequent layer. Therefore, the final deployed model does not require the SNN platform to support graded spikes and remains fully compatible with most event-driven neuromorphic hardware. We will make this clearer in the revised manuscript to avoid potential confusion.
>
> > Eq (6) why is the output of the i-th neuron, s_i, a vector? Also, y_hat should have both i and k indices.
>
> We thank the reviewer for these helpful observations.
>
> In Equation (6), $s_i(x_k)$ is treated as a vector because it represents the spike outputs of the i-th neuron across all time steps when processing the k-th input sample. Each neuron's output is a sequence of binary spike events over time. We will add description in the manuscript.
>
> We agree that $\hat{y}_i$ should depend on the input sample and should include both the neuron index i and the input index k. Thank you for pointing this out. We will correct the notation in the revised version to ensure clarity and consistency.
>
> > There 1: Why is it derivative in (8) and gradient in (9)? Are (8,9) saying the same thing? Loose index j in (8).
>
> Thank you for pointing this out. We apologize for the confusion caused by Equations (8) and (9), and we will revise them in the manuscript to improve clarity. Specifically, the use of partial derivative notation in (8) and gradient notation in (9) was inconsistent. As $w_{ij}$ is a scaler, we will revise the Equation (9) to derivative.
> Also index i for $s_i[t]$ in Equation (8) is incorrect and may lead to confusion. Since $w_{ij}$ denotes the weight from neuron j to neuron i, the corresponding spike output should be indexed by j, not i. We will correct this in the revised version by replacing it.
>
> We appreciate the reviewer’s efforts for improving our manuscript and we will carefully check the equations and explanations in the manuscript to improve clarity and mathematical rigor.
>
> > Zero gradient is not exactly the same as a saddle point. Maybe give a hint of the proof in the main text.
>
> We sincerely thank the reviewer for pointing out this critical issue. We acknowledge that our use of the term "saddle point" in this context was not only imprecise but incorrect. The proper terminology should be critical point, defined as any point where the gradient vanishes.
>
> We agree that the misuse of terminology undermines the rigor of the presentation. We deeply apologize for this mistake and will correct it in the revised manuscript by replacing "saddle point" with "critical point" and including a concise explanation of the result in the main text for clarity and correctness. However, we would like to point out that our argument remains valid, that the extreme sparsity of spike activity leads to vanishing gradients and thus impedes learning with gradient-based optimization methods.
>
> > Line 179: subscript ()_t is never used
>
> Thanks for your suggestion. In Equation (10), we use $\theta_{i,t}$ to denote the temporary value of the neuron threshold during the iterative update. However, we agree that this equation is confusing and unnecessary (as also pointed out by Reviewer 7Qnw). We will remove it and replace it with a clearer narrative explanation to improve clarity.
>
> > Line 196 BPTT requires memory of previous time states? Please clarify.
>
> Thanks for pointing it out. Yes, training spiking neural networks with Backpropagation Through Time (BPTT)[1] does require maintaining memory of the neuronal states across all time steps during the forward pass. This includes membrane potentials, spike outputs, and other internal variables such as the threshold values in our case. BPTT unrolls the SNN over time and computes gradients by propagating errors backward through each time step. Therefore, for accurate gradient computation, it is necessary to store all intermediate states over the full simulation window.
>
> We will clarify this detail in the revised manuscript.
>
> > Doesn’t it make sense to do overall evaluation first, then ablation?
>
> Thanks for your suggestion. In the revised version, we will restructure the experiments by first presenting the main performance results and then followed by the ablation study.
>
> > Line 260 what’s synaptic operation numbers?
>
> Thanks for your suggestion. The synaptic operation number (SOP) refers to the metric that calculate the total number of computational events. Specifically, one SOP is counted whenever a spike is transmitted from a presynaptic neuron to a postsynaptic neuron through a weighted connection. This metric is widely used in SNN literature as an estimate of the computational cost and energy consumption of a model [2, 3]. We will clarify this definition in the revised manuscript for better understanding.
>
> > Line 271: if the threshold \theta is constant, what’s the difference if the signal is 0-1 or 0-\theta?
>
> Thank you for the insightful question. As noted in our earlier response regarding the "graded spike" clarification, there is indeed no difference during inference when $\theta$ is constant, since the scaling factor can be absorbed into the weights of the following layer.
>
> However, during training, $\theta$ is not constant. It is treated as a learnable parameter and is dynamically adapted over time. In addition, we apply regularization on $\theta$ to suppress spike activity. This design allows $\theta$ to act as a learnable scaling factor, which helps to maintain stable gradient and effective representation learning, while also encouraging sparsity through threshold value growth. We will revise the manuscript to clarify this.
>
> > Line 272: what’s synaptic output feature?
>
> We apologize that the phrase "synaptic output feature" is imprecise and may cause confusion. What we intended to describe is the current-based output used in the AT-LIF neuron, where the output is scaled by the threshold (0 or θ) instead of the spike output (0 or 1). We will rephrase the term "synaptic output feature" by "current-based output" to avoid ambiguity.
>
> > Line 281: what is SOTA?
>
> “SOTA” is the abbreviation for State-of-the-Art. In our manuscript, it refers to the most advanced existing methods in the literature at the time of writing. We will replace the abbreviation to ensure clarity for all readers.
>
> > Table 2: some of the best AT-LIF results are with \lambda_s=0, i.e. no sparsity. Please comment. What is the column T?
>
> Thanks for pointing it out. To clarify:
>
> $\lambda_s$ is the coefficient for direct activation regularization, which penalizes spike activity in the loss function. When $\lambda_s$ = 0, it means there is no explicit regularization on spike activity. $\lambda_t$ on the other hand, is the coefficient for threshold adaptation regularization introduced in our AT-LIF neuron model. It indirectly controls spike sparsity by increasing the firing threshold during training. Even when $\lambda_s$ = 0, the presence of a non-zero $\lambda_t$ enables our model to suppress spike activity effectively.
>
> Regarding the column T, it represents the number of discrete simulation time-steps used. In the context of spiking neural networks using the discrete-time Leaky Integrate-and-Fire model, T defines how many times the neuron dynamics are updated over time in response to the given input. In table 2, a fixed T is used across models for fair comparison.
>
> We will revise the manuscript to include these clarifications.
>
> > Please give more details how your method is integrated with other methods in the experiments.
>
> We thank the reviewer for this question and are happy to provide more details.
>
> In our experiments, we combine our proposed AT-LIF method with STDS [4], which is a state-of-the-art weight pruning algorithm designed for SNNs. STDS operates by reparameterizing the weights and progressively eliminating those with small absolute values during training, leading to a sparse connectivity pattern.
>
> Our integration is straightforward: we replace the standard LIF neurons with AT-LIF neurons throughout the network, and apply threshold adaptation as described in our method. Meanwhile, the STDS pruning algorithm is used to train the synaptic weights and induce weight sparsity. This combination allows us to simultaneously achieve both activity sparsity and weight sparsity, through STDS's connection-level pruning. In practice, AT-LIF focuses on reducing spike activity, while STDS reduces spatial connections.
>
> We will clarify this integration process in the revised manuscript to improve clarity and reproducibility. If there are any remaining questions about how the two methods are combined, please feel free to let us know.
>
> **References**
>
> [1] Wu, Yujie, et al. "Spatio-temporal backpropagation for training high-performance spiking neural networks." Frontiers in Neuroscience, 2018.
>
> [2] Hu, Yangfan, Huajin Tang, and Gang Pan. "Spiking deep residual networks." TNNLS, 2021.
>
> [3] Shi, Xinyu, et al. "Towards energy efficient spiking neural networks: An unstructured pruning framework." ICLR, 2024.
>
> [4] Chen, Yanqi, et al. "State transition of dendritic spines improves learning of sparse spiking neural networks." ICML, 2022.

---

> > ### Comment · Reviewer_CMEG · 2025-08-06
> >
> > Thanks for your thorough attention to my comments. The responses are to my satisfaction.
> > As a last minor point, in the equation13 the subscript in \lambda_t may be confused with the index in summation.

---

> > > ### Author Response · Authors · 2025-08-07
> > >
> > > We sincerely thank the reviewer for the thoughtful feedback. We appreciate your attention to this detail and agree that the subscript in Equation (13) may cause confusion. To avoid this ambiguity, we will revise the notation and replace $\lambda_t$ with $\lambda_{thr}$ in the final version of the manuscript.

---

> > > > ### Comment · Reviewer_CMEG · 2025-08-08
> > > >
> > > > Thank you for attention to my comments

---

### Official Review · Reviewer_364k · 2025-06-30

**Clarity:** 3
**Significance:** 3
**Originality:** 3
**Rating:** 5
**Confidence:** 4

**Summary:**

This paper investigates the problem of high neuronal activity in SNNs.
The authors identify the limitations of current activity regularization methods and propose a new neuron model, AT-LIF, alongside a threshold adaptation mechanism, to suppress spike activity during training.
Notably, the method achieves promising results on ImageNet with a very low average firing rate and a compact model size.

**Questions:**

1. Has the proposed method been evaluated on real neuromorphic hardware (e.g., Loihi, Speck)?
2. Does the introduction of threshold adaptation increase training complexity or require additional hyperparameter tuning?
3. Can the authors discuss whether the method can be applied to other domains (e.g., NLP, speech) or large architecture like spiking Transformers?

**Ethical Concerns:**

["NO or VERY MINOR ethics concerns only"]

**Final Justification:**

My concerns have been addressed.

**Limitations:**

Yes.

**Quality:**

4

**Strengths And Weaknesses:**

## Strengths
1. I think the efficiency gains are clear. The method achieves a substantial reduction in firing rate and synaptic operations.
2. The authors propose a modified neuron model (AT-LIF) and threshold adaptation to promote sparsity in SNNs, addressing a core inefficiency in SNN deployment.
3. I think the compatibility with pruning methods suggests broader applicability and system-level optimization potential.

## Weaknesses
No obvious weaknesses.

---

> ### Author Rebuttal · Authors · 2025-07-29
>
> # Response to Reviewer 364K
> > Has the proposed method been evaluated on real neuromorphic hardware (e.g., Loihi, Speck)?
>
> Thanks for your suggestion. We acknowledge that our current work has not yet included an evaluation on real neuromorphic hardware platforms. As discussed in the paper, our study primarily focuses on the algorithmic development and analysis of spike activity pruning at the model level. To estimate the efficiency benefits and potential for hardware deployment, we instead report metrics such as average firing rates and synaptic operation counts (SOPs), following the standard practices in recent literature. These metrics are widely used to estimate energy and computational cost on neuromorphic systems [2,3], and we believe they provide a meaningful estimation of the performance gain achieved by our method.
>
> Nonetheless, we fully agree that empirical validation on real hardware is an important next step. As we discussed in the section *Limitations*, we plan to explore such evaluations in future work, which could offer deeper insights into the real-world applicability of our proposed approach.
>
> > Does the introduction of threshold adaptation increase training complexity or require additional hyperparameter tuning?
>
> We thank the reviewer for raising this insightful question. Regarding training complexity, we would like to clarify that the introduction of the threshold adaptation mechanism in AT-LIF neurons does not significantly increase the computational burden. Specifically, the threshold adaptation involves a lightweight update to the threshold value at each iteration. Its computational cost is minimal compared to the standard backpropagation process, which involves updating high-dimensional weight matrices. In our implementation on single RTX 4090, we observed no noticeable overhead during training on GPUs. For example, using the ResNet-20 architecture with 32×32 input resolution on CIFAR-10, the vanilla model requires approximately 40.1 seconds per epoch, while the AT-LIF model takes about 40.5 seconds per epoch, which is a negligible increase.
>
> As for hyperparameter tuning, we acknowledge that our method introduces an additional coefficient $\lambda_t$, which controls the strength of the threshold regularization. In practice, we use standard strategies such as grid search or binary search over a small range. However, we would like to emphaize that unlike weight pruning, it is hard to directly control the desired sparsity of the spike activity, as different input sample triggers distinct pattern of spikes, resulting in different activation sparsity. Thus empirically setting the regularization coefficient for the penalty term is a more practical and feasible solution. Moreover, in our method, the convergence of the model is more robust to extreme values of the regularization coefficient. For example, with $\lambda_t=1e-3$, ResNet-20 on CIFAR-10 achieves 92.05% accuracy with 61.7M SOPs. Even when $\lambda_t$ is increased to $1e-1$, the model is still able to converge, achieving 45.52% accuracy with only 9.19M SOPs and 0.007 average firing rate. In contrast, for direct activation regularization, the model achieves 91.31% accuracy with 57.26M SOPs when $\lambda_s = 1e-2$, but fails to converge when $\lambda_s \geq 0.03$. This highlights the greater stability and flexibility of our method during hyper-parameter selection.
>
> We will incorporate this discussion into the revised manuscript for better clarity and explicitly mention this in the *Limitations* section.
>
> > Can the authors discuss whether the method can be applied to other domains (e.g., NLP, speech) or large architecture like spiking Transformers?
>
> We thank the reviewer for this insightful question. We believe that our proposed method is broadly applicable across domains and architectures, including tasks in computer vision and natural language processing. Since the AT-LIF operates at the neuron level and does not rely on task-specific or architecture-specific assumptions, it can be easily integrated into most spiking neural networks.
>
> We extended our method to a spiking Transformer architecture (QKFormer[1]) and conducted experiments on the CIFAR-10 dataset. The AT-LIF neuron was seamlessly incorporated into the attention blocks, and we observed that the method remained effective in reducing spike activity while maintaining competitive accuracy. As shown in the table below, our method achieved a substantial reduction in average firing rate (from 0.106 to 0.061) and synaptic operation count (SOPs reduced from 444.9M to 281.3M), with only a 0.59% drop in accuracy compared to the vanilla model. These results suggest that the AT-LIF mechanism continues to scale well to Transformer-based architectures, offering significant gains in efficiency with minimal compromise in performance.
>
> |          | Accuracy | SOPs (M) | Avg. FR. |
> | -------- | -------- | -------- | -------- |
> | Vanilla  | 89.86    | 444.9    |  0.106   |
> | AT-LIF   | 89.27    | 281.3    |  0.061   |
>
> While we have not yet applied the method to NLP or speech tasks due to limited rebuttal time, we believe the core idea is generalizable and plan to explore these directions in future work. We will include these results and expand the discussion in the revised manuscript.
>
> **Reference**
>
> [1] Zhou, Chenlin, et al. "Qkformer: Hierarchical spiking transformer using qk attention." NeurIPS, 2024.
>
> [2] Hu, Yangfan, Huajin Tang, and Gang Pan. "Spiking deep residual networks." TNNLS, 2021.
>
> [3] Shi, Xinyu, et al. "Towards energy efficient spiking neural networks: An unstructured pruning framework." ICLR, 2024.

---

> ### Comment · Reviewer_364k · 2025-08-01
>
> Thanks for your reply, my concerns have been addressed. I decide to raise my score.

---

### Official Review · Reviewer_CCTV · 2025-07-01

**Clarity:** 3
**Significance:** 2
**Originality:** 3
**Rating:** 5
**Confidence:** 5

**Summary:**

Motivated by the biological principle of sparse neural coding and the need for efficient computation, the authors propose AT-LIF, an adaptive threshold leaky integrate-and-fire neuron model. AT-LIF dynamically adjusts neuron firing thresholds to suppress spike activity while maintaining output fidelity, effectively overcoming optimization conflicts and gradient vanishing issues in training sparse SNNs. The method is evaluated on CIFAR-10, DVS-CIFAR10 ImageNet across different architectures, demonstrating significant reductions in average firing rates and synaptic operations, while preserving classification accuracy in some cases. These results highlight the method's potential for building scalable and energy-efficient SNNs.

**Questions:**

None.

**Ethical Concerns:**

["NO or VERY MINOR ethics concerns only"]

**Final Justification:**

The authors have well addressed all my concerns. The proposed method can significantly reduce average firing rates and synaptic operations while preserving classification accuracy, which is promising for neuromorphic chips. I decide to increase my score to 5.

**Limitations:**

None.

**Paper Formatting Concerns:**

None.

**Quality:**

3

**Strengths And Weaknesses:**

Strengths:
The paper introduces a novel activity-pruning method inspired by sparse coding in the brain. This direction departs from conventional weight-based pruning and aligns well with the natural efficiency of neural systems.
The proposed AT-LIF neuron model is both innovative and effective. It addresses two major challenges in training sparse SNNs, the gradient vanishing and optimization conflict, through adaptive thresholding and current-based output mechanisms. The approach demonstrates compatibility with existing weight-pruning techniques. The combined use of AT-LIF with methods like STDS enables both activity and weights sparsity, significantly improving computational efficiency.

Weaknesses:
The method still exhibits a clear trade-off between accuracy and sparsity. As shown in the experiments, increasing the threshold adaptation coefficient to induce lower spike activity leads to performance degradation.

---

> ### Author Rebuttal · Authors · 2025-07-29
>
> # Response to Reviewer CCTV
> > The method still exhibits a clear trade-off between accuracy and sparsity. As shown in the experiments, increasing the threshold adaptation coefficient to induce lower spike activity leads to performance degradation.
>
> We thank the reviewer for highlighting this point. We agree that a trade-off between model accuracy and spike activity sparsity still exists, particularly when the threshold adaptation coefficient $\lambda_t$ is increased aggressively. In fact, suppressing spike activity toward near-zero firing rates inevitably constrains the network’s expressiveness and lead to performance degradation. However, we would like to emphasize that our method provides a way to balance such trade-off more gracefully than direct activity regularization methods.
>
> We compare the AT-LIF method with more existing spike activity pruning methods, including L1 regularization on the firing rate (L1 Reg [1]), Hoyer regularization over the output spike (Hoyer Reg [2]) and the baseline method presented in the manuscript (AR).
>
> As shown in both Table 2 in the manuscript and the below table, AT-LIF achieves competitive accuracy under moderate sparsity constraints and demonstrates better performance than conventional activation regularization under similar average firing rates.
>
> **ResNet-20, CIFAR-10**
> |          | Accuracy | SOPs (M) | Avg. FR. |
> | -------- | -------- | -------- | -------- |
> | L1 Reg [1]    | 91.27    | 60.21    |  0.062   |
> | Hoyer Reg [2] | 88.62    | 76.79    |  0.081   |
> | AR [Baseline] | 91.31    | 57.26    |  0.060   |
> | AT-LIF [Ours] | 91.06    | 48.85    |  0.043   |
>
> Nonetheless, we acknowledge that this trade-off remains a fundamental challenge, especially for extreme sparse models. We will explicitly add this discussion to the *Limitations* section of the revised manuscript.
>
> **References**
>
> [1] Deng, Lei, et al. "Comprehensive snn compression using admm optimization and activity regularization." TNNLS, 2021.
>
> [2] Narduzzi, Simon, et al. "Optimizing the consumption of spiking neural networks with activity regularization." ICASSP, 2022.

---

### Decision · Program_Chairs · 2025-09-17

**Decision:**

Accept (poster)

**Comment:**

This paper introduces AT-LIF, an adaptive threshold leaky integrate-and-fire neuron model designed to reduce spiking activity in spiking neural networks (SNNs) while preserving task performance. The approach addresses key challenges in training sparse SNNs, including gradient vanishing and optimization conflicts, through threshold adaptation and current-based outputs. Empirical results across CIFAR-10, DVS-CIFAR10, and ImageNet show significant reductions in firing rate and synaptic operations, often with minimal accuracy loss, and demonstrate compatibility with weight pruning methods.

Reviewers agree that the contribution is technically solid, original, and relevant for energy-efficient neuromorphic computing. While some noted issues with clarity, limited comparisons in the initial submission, and the inherent trade-off between accuracy and sparsity, the authors’ rebuttal effectively addressed these concerns with additional results and clarifications.

Overall, the consensus is that this is a strong and promising contribution to efficient SNN training, with clear potential impact on scalable neuromorphic systems.